# HIV prevalence and awareness among adults presenting for enrolment into a study of people at risk for HIV in Kisumu County, Western Kenya

**Valentine Sing'oei[1,2], Chiaka Nwoga[3,4], Adam Yates[3,4], John Owuoth[1,2], June Otieno[2], Erica Broach[3,4], Qun Li[3,4], Zebiba Hassen[3,4], Michelle Imbach[3,4], Mark Milazzo[3,4], Tsedal Mebrahtu[3,4], Merlin L. Robb[4], Julie A. Ake[3,4], Christina S. Polyak[3,4], Trevor A. Crowell[3,4]\*, on behalf of the RV393 Study Group[¶]**

1 HJF Medical Research International, Kisumu, Kenya, 2 United States Army Medical Research Directorate-Africa, Nairobi, Kenya, 3 U.S. Military HIV Research Program, Walter Reed Army Institute of Research, Silver Spring, MD, United States of America, 4 Henry M. Jackson Foundation for the Advancement of Military Medicine, Bethesda, MD, United States of America

¶ Membership of the RV393 Study Group is provided in the Acknowledgments.
\* tcrowell@hivresearch.org

**Data Availability Statement:** The dataset supporting the conclusions of this article is

## Abstract

### Introduction

Despite declines in new HIV diagnoses both globally and in Kenya, parts of Western Kenya still report high HIV prevalence and incidence. We evaluated HIV prevalence to inform the development of policies for strategic and targeted HIV prevention interventions.

### Methods

Adult participants aged 18–35 years were recruited in Kisumu County and screened for HIV for a prospective HIV incidence cohort. Questionnaires assessed HIV-associated risk behaviors. Participants who tested positive for HIV were disaggregated into groups based on prior knowledge of their HIV status: previously-diagnosed and newly-diagnosed. In separate analyses by prior knowledge, robust Poisson regression was used to estimate prevalence ratios (PRs) and 95% confidence intervals (CIs) for factors potentially associated with a positive HIV test in each group, as compared to participants without HIV.

### Results

Of 1059 participants tested for HIV, 196 (18.5%) had a positive HIV test. Among PLWH, 78 (39.8%) were newly diagnosed with HIV at screening. After adjusting for other variables, previously-diagnosed HIV was more common among females than males (PR 2.70, 95%CI 1.69–4.28), but there was no observed sex difference in newly-diagnosed HIV prevalence (PR 1.05, 95%CI 0.65–1.69). Previously-diagnosed HIV was also more common among people reporting consistent use of condoms with primary sexual partners as compared to inconsistent condom use (PR 3.19, 95%CI 2.09–4.86), but newly-diagnosed HIV was not

available in the Harvard Dataverse repository, https://doi.org/10.7910/DVN/7OBISJ

**Funding:** This work was supported by cooperative agreements (W81XWH-11-2-0174; W81XWH-18-2-0040) between the Henry M. Jackson Foundation for the Advancement of Military Medicine, Inc., and the U.S. Department of Defense. This research was funded, in part, by the U.S. National Institute of Allergy and Infectious Diseases. The funders had no role in study design, data collection and analysis, decision to publish, or preparation of the manuscript.

**Competing interests:** The authors have declared that no competing interests exist.

**Abbreviations:** MSM, men who have sex with men; PLWH, people living with HIV; PLWOH, people living without HIV; UNAIDS, Joint United Nations Programme on HIV and AIDS; VCT, HIV voluntary counseling and testing.

associated with such a difference between consistent and inconsistent condom use (PR 0.73, 95%CI 0.25–2.10).

## Conclusion

Prevalence of newly-diagnosed HIV was high, at approximately 8% of participants, and not statistically different between genders, highlighting the need for improved HIV case finding regardless of sex. The higher prevalence of previously-diagnosed HIV in female participants may reflect higher rates of HIV testing through more encounters with the healthcare system. Higher prevalence of consistent condom use amongst those previously-diagnosed suggests behavioral change to reduce HIV transmission, a potential benefit of policies to facilitate earlier HIV diagnosis.

## Introduction

HIV voluntary counseling and testing (VCT) is an essential gateway to HIV prevention, treatment, care and supportive services. In the past decade, there has been a significant decline of HIV incidence globally [1]. Factors contributing to this reduction include scaling up of innovative HIV case finding strategies such as assisted partner notification services [2], index case testing, extended working hours and demand creation using the media [3] leading to increased treatment coverage and viral suppression with subsequent reduction of HIV-related mortality. However, rising numbers of new HIV cases are being observed in some populations, partially driven by transmission from undiagnosed cases and known cases who have not initiated suppressive antiretroviral therapy (ART) [4].

Globally, there are an estimated 39 million people living with HIV (PLWH), 29.8 million of whom are on ART, which leaves a treatment gap of 9.2 million people [5]. Sub-Saharan Africa accounts for only about 13% of the world population, but carries a disproportionate burden of HIV, accounting for greater than 70% of PLWH [6]. Globally in 2022, 86% of people living with HIV (PLWH) knew their status, of which 88% were accessing treatment [7]; this means 14% were unaware of their status and another 12% knew their status but were not on ART, highlighting the challenge of achieving UNAIDS 95-95-95 goals in some regions. In the African region, 90% of PLWH knew their status, of which 82% were accessing treatment [5]; this means 10% were unaware of their status and another 18% knew their status but were not on ART. Most new cases of HIV occur through sexual transmission and more commonly from persons unaware of their HIV status as people who are aware are more likely to adopt prevention methods such as condom use and are more likely to be on suppressive ART [8, 9].

Kenya ranks among the top fifteen countries in the world with the highest prevalence of HIV, approximately 3.7% in adults aged 15 to 49 years with prevalence in women twice as high compared to men [10, 11]. Kenya has a mixed HIV epidemic with geographic and population variations. Approximately 30% of new HIV diagnoses occur in key and priority populations including men who have sex with men (MSM), female sex workers, and adolescent girls and young women. Key populations constitute a small proportion of the general population but are at an elevated risk of acquiring HIV in part due to social exclusion, mental health perturbations, discrimination, and decreased engagement with healthcare services [8, 12–16]. In Kenya, as has occurred worldwide, there has been a notable decline in new HIV cases due to scaling up ART coverage and implementing strategic combination prevention interventions.

However, these gains have not been achieved uniformly across the country. For example, some counties in Western Kenya report an HIV prevalence about three times the national prevalence. Kisumu County has the second highest prevalence in the country at 17.4%, with an incidence of 6.9 cases per 1,000 person-years [17]. Lack of awareness of HIV status, failures in linkage to ART, and behavioral factors are key potential drivers of HIV transmission.

This study assessed HIV prevalence and HIV status awareness among potential participants in an HIV incidence study in Kisumu County, with the aim of informing the development of policies for strategic and targeted HIV prevention and testing interventions.

## Methods

### Study participants

Data for these cross-sectional analyses were obtained from the screening visit to assess eligibility for enrollment into RV393, a prospective observational study to estimate HIV prevalence and incidence in order to inform the feasibility of potential future efficacy trials of candidate preventive HIV vaccines. From January 27, 2017 to May 17, 2018, male and female participants were enrolled who were aged 18–35 years and resided in Kisumu County.

### Recruitment strategy

Community support was promoted through engagement of a community advisory board and communication with leaders, established organizations, County, Sub-County and local Ministry of Health leadership. We identified organizations in each sector that could serve as recruitment and referral sites, leveraging these to conduct focus group discussions with village chiefs and other community leaders. Community health workers coordinated small group or informal gatherings of individuals from communities known to be particularly vulnerable to HIV, such as female sex workers, fishmongers, young women, and MSM. Direct, targeted recruitment was conducted at neighborhood locations known to be venues for risk-taking behaviors, such as bars, markets, fishing villages, military barracks, beauty salons, and social events. Recruitment also targeted secondary night schools and universities in and around the study site, as well as HIV VCT centers.

### Study procedures

At screening for study eligibility, a complete medical history and physical examination was performed, and trained site staff administered questionnaires. A social history was collected that included documentation of any history of alcohol abuse, defined as having more than five alcoholic drinks in one day and/or causing harm while intoxicated. Information on sexual activities and other behaviors associated with HIV risk was collected such as consistent condom use, defined as self-reported 'always' using condoms with primary and/or secondary sexual partners. Transactional sex was defined as having sex with someone in exchange for money, goods, gifts, or favors in the last 3 months. Employment was defined as self-report of any occupation. The questionnaire also included information about age at first sex; sexual behaviors in the last three months (total number of partners; presence of a primary partner defined as a boyfriend, girlfriend or spouse; and presence of a secondary partner); and self-reported history of being diagnosed with a sexually transmitted infection (STI) by a health professional in last three months. Because participants were not asked to self-identify their sexual orientation/identity, 'men who had sex with men' (MSM) is defined using the variable 'partner gender', which asked participants if their sexual partners in the prior 3 months were 'male', 'female' or 'both genders'; male participants who indicated having partners that were 'male' or

'both' genders were considered 'MSM'. The variable 'anal sex with men' was collected as a sexual behavior of interest; participants who indicated having engaged in anal sex with men were coded as 'yes' while all participants who did not indicate this activity are coded as 'no'. A participant's prior knowledge of their HIV status was determined through the survey item for self-assessed HIV risk ("How would you classify your risk for getting infected with HIV?"), which included a response of "I know I am HIV positive."

HIV status was determined using the First Response® Rapid HIV 1–2–0 card tests (Premier Medical Corporation, Maharashtra, India). HIV ELISA via the GenScreen ULTRA HIV Ag-Ab Combo Kit was used for inconclusive rapid tests and HIV was confirmed with the Geenius HIV-1/2 Supplemental Assay (Bio-Rad Laboratories, Hercules, CA). Participants were informed of their HIV diagnosis after the collection of all other study evaluations and questionnaires.

## Ethics approval and consent to participate

This study was reviewed and approved by the Kenya Medical Research Institute Scientific and Ethical Review Unit, Nairobi, Kenya and the Walter Reed Army Institute of Research Institutional Review Board, Silver Spring, Maryland USA. The protocol used for this study complied with International Conference on Harmonization Good Clinical Practice guidelines and was conducted in accordance with the principles described in the Nuremberg Code and the Belmont Report including all federal regulations regarding the protection of human participants as described in 32 CFR 219 and Army Regulation 70–25. All participants and impartial witnesses of illiterate participants provided written informed consent before screening for study eligibility.

## Statistical analysis

These analyses included individuals who completed a screening visit to assess eligibility for inclusion in the longitudinal cohort, completed a behavioral questionnaire, self-reported that they had experienced their sexual debut, and had available HIV test results. As knowledge of living with HIV may have a significant effect on associated behaviors, participants who tested positive for HIV at screening for study eligibility (i.e., PLWH) were disaggregated into two groups based on their prior knowledge of their HIV status: previously-diagnosed or newly-diagnosed. Independent analyses were conducted for each group, with the same pool of participants without HIV serving as the comparator group in each model.

Demographic and distributional characteristics were stratified by participant knowledge of their HIV status and summarized using frequencies. Chi-square tests and t-tests were used to assess the association between HIV status among categorical and continuous covariates, respectively. Population prevalence for each prior knowledge group included PLWH of a given knowledge status in the numerator and added all participants without HIV for the denominator. Univariable and multivariable models were constructed to identify significant covariate associations with HIV status for each knowledge group independently using complete-case analyses. Because of the strong correlation between 'MSM' and 'anal sex with men' categories, only one variable can be included in the regression models. As MSM is an identity while 'anal sex with men' is an exposure risk factor, we include the latter in regression analyses. Participants with missing covariate data were excluded from analyses using listwise deletion of participants from the regression models; missingness due to listwise deletion was 3.9% for models of previously undiagnosed HIV and 4.0% for previously-diagnosed. In order to directly model associations, prevalence models were constructed as Poisson models with robust sandwich estimators of standard errors [15]. The models were run separately for each prior

knowledge category, with 'events' defined as participants who tested positive for HIV at screening and 'non-event' defined as not testing positive at the time of screening. To facilitate comparisons and control for potential confounding, variables that were significant in the uni-variable models with a p<0.05 were included in multivariable models. Statistical analyses were conducted in SAS v9.4.

## Results

### Characteristics of study population

Of the 1072 Kenyan adults who were consented, 1068 reported that they had experienced their sexual debut, and 1059 with available behavioral survey data and HIV testing results were included in these analyses. Of these, 540 (51.0%) were females and the median age of included participants was 25 years (interquartile range [IQR] 22–29). The number of PLWH was 196, which reflected an overall HIV prevalence of 18.5%, with the highest prevalence being observed in Kisumu City and the areas immediately to its west ([Fig 1]). As compared to people living without HIV (PLWOH), PLWH were more likely to be female (73.0% vs. 46.0%, p<0.001) and tended to be older (median 29 [IQR 26–31] vs. 24 [21–28] years, p<0.001). PLWH and PLWOH reported a similar total number of sexual partners in the preceding three months (median 3 [IQR 1–7] and 3 [IQR 2–6] partners, respectively, p = 0.128).

As compared to PLWOH, participants who were previously-diagnosed with HIV were more often females (82.2% vs. 46.0%, p<0.001), tended to be older (median 28 [IQR 26–31]

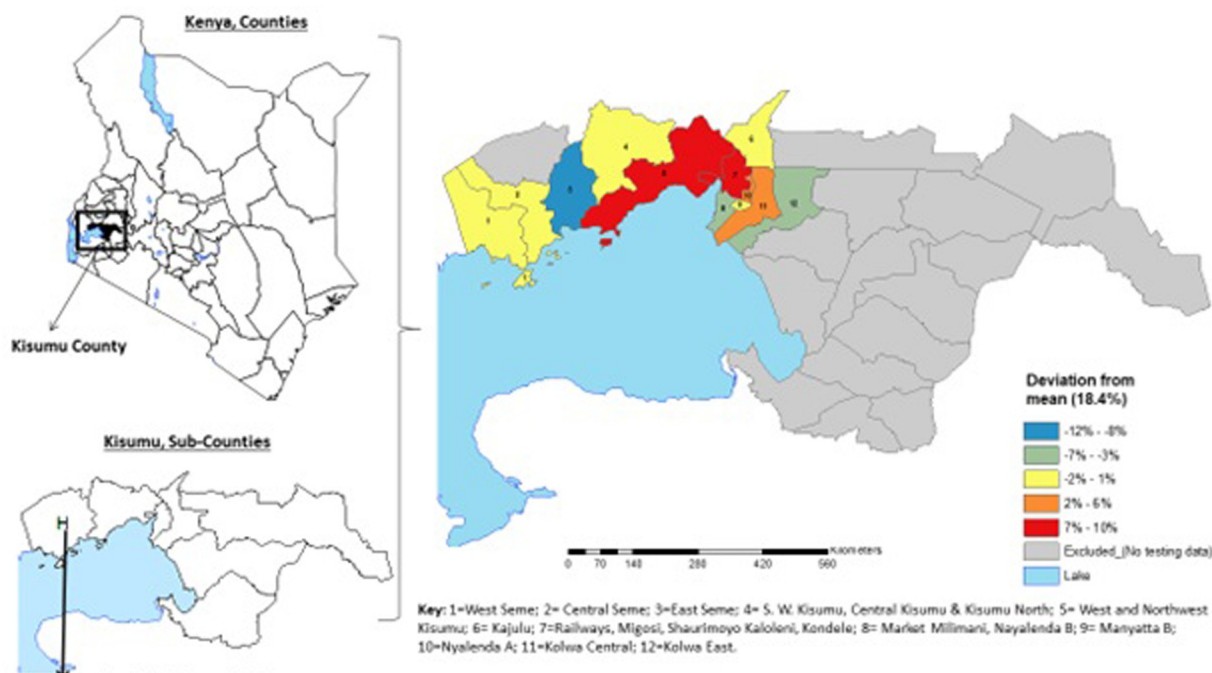

**Fig 1. HIV prevalence deviation from the mean in Kisumu, Kenya.** Fig 1 depicts the subregional deviation in HIV prevalence from overall point prevalence [18] PLWH resided in 4 sub-regions located centrally in the area around and stretching westward and eastward of Kisumu City. The sub-region encompassing Kisumu City (sub-region 7) and that stretching westward (sub-region 5) had the highest numbers with an overall deviation of between 7% and 10% from the overall HIV prevalence. Among the 196 PLWH, 118 (60.2%) were previously diagnosed as living with HIV while 78 (39.8%) were previously undiagnosed. Participant residential wards were combined due to low N in some areas. Gray areas represent areas of Kisumu county wherein data collection for the RV393 study did not occur. The shapefile was retrieved from Stanford Digital Repository [Map Maker Ltd. (2007)]. Sub-locations, Kenya, 2007 Map Maker, Ltd. Available at: https://purl.stanford.edu/cb941rz5380. This figure was generated using the ArcGIS Version 10.3.1 (Environmental Systems Research Institute, Red-lands, CA, USA) software.

vs. 24 [IQR 21–28] years, p<0.001), and reported lower monthly income (median 7000 [IQR 4000–15000] vs. 9000 [5000–15000] Kenyan shillings, p = 0.014). The previously-diagnosed population also differed significantly from PLWOH in marital status, education, proportion reporting secondary sexual partners, and consistency of condom use with secondary partners (all p<0.001, Table 1).

As compared to PLWOH, participants with newly-diagnosed HIV were more often females (59.0% vs. 41.0%, p = 0.029) and tended to be older (median 29 [IQR 26–31] vs. 24 [21–28] years, p<0.001). Participants with newly-diagnosed HIV reported similar monthly income to PLWOH (median 9000 [IQR 4000–15000] vs. 9000 [5000–15000] Kenyan shillings, p = 0.897). The newly-diagnosed population also differed significantly from PLWOH in marital status and education (all p<0.001, Table 1).

Among PLWOH, 63.2% had been tested for HIV within 6 months prior to their screening visit for this study. Among participants with previously-diagnosed HIV, 30.2% reported that their last HIV test was within the preceding 6 months and 54.3% indicated that their last HIV test was over one year prior. Among participants with newly-diagnosed HIV, 28.2% reported that their last HIV test was within the preceding 6 months and 41.0% reported having their last HIV test over one year prior.

### Factors associated with previously-diagnosed HIV

Participants who were diagnosed with HIV prior to screening for study eligibility were compared to PLWOH in a multivariable model (Table 2). Previously-diagnosed HIV was more common among female participants as compared to males (PR 2.70, 95%CI 1.69–4.28) and increased with age (25–29 years: (PR 2.62 [95% CI 1.51–4.57], 30+ years: PR 3.70 [95%CI 2.70–6.42], as compared to 18–24 years). Compared to individuals who did not consistently use a condom with their primary sexual partner, previously-diagnosed HIV was over 3 times more common among individuals with consistent condom use (PR 3.19, 95%CI 2.09–4.86). Previously-diagnosed HIV tended to be more common among married individuals as compared to single individuals, although the difference was not statistically significant (PR 1.30, 95%CI 0.82–2.07). Previously-diagnosed HIV tended to be least common amongst the wealthiest quintile income group, and the comparison to the bottom quintile was statistically significant (PR 0.52, 95%CI 0.29–0.91). Additionally, previously-diagnosed HIV was 2 times more common among individuals with an older age of sexual debut (PR 2.36, 95%CI 1.03–5.39). The prevalence of previously-diagnosed HIV did not vary significantly by marital status or consistency of condom use with secondary sexual partners in the multivariable model.

### Factors associated with newly-diagnosed HIV

Participants who were newly-diagnosed with HIV at their screening visit for study eligibility were compared to PLWOH in a multivariable model (Table 2). The prevalence of newly-diagnosed HIV was significantly higher in the older age categories of ages 25–29 years (PR 3.69 [95%CI 1.81–7.52]) and ages 30+ years (PR 5.68 [95%CI 2.62–12.31]) compared to those aged 18–24 years. The prevalence of newly diagnosed HIV did not vary by sex, marital status, age at sexual debut, or consistency of condom use with primary or secondary sexual partners in the multivariable model.

## Discussion

The Joint United Nations Programme on HIV and AIDS (UNAIDS) estimates that about 6 million PLWH globally do not know their HIV status[1, 19] and this group contributes substantially to new HIV transmissions. In our study, more than one-third of PLWH were not

**Table 1. Demographic and descriptive characteristics of adults presenting for enrollment into an HIV incidence study in Kisumu, Kenya, stratified by knowledge of their HIV status.**

| Parameter | Category | PLWOH, n(%) Total N = 863 | Previously diagnosed HIV status, n(%) Total N = 981 | With HIV N = 118 | P-value (Previously diagnosed vs. PLWOH) | Newly diagnosed HIV status, n(%) Total N = 941 | With HIV N = 78 | P-value (Newly diagnosed vs. PLWOH |
|---|---|---|---|---|---|---|---|---|
| **Sex** | Male | 466(54.0) | 487 (49.6) | 21(17.8) | < .0001 | 498 (52.9) | 32(41.0) | **0.0279** |
| | Female | 397(46.0) | 494 (50.4) | 97(82.2) | | 443 (47.1) | 46(59.0) | |
| **Age** | 18 to 24 years | 468(54.2) | 489 (49.8) | 21(17.8) | < .0001 | 481 (51.1) | 13(16.7) | < .0001 |
| | 25 to 29 years | 250(29.0) | 298 (30.4) | 48(40.7) | | 282 (30.0) | 32(41.0) | |
| | ≥30 years | 145(16.8) | 194 (19.8) | 49(41.5) | | 178 (18.9) | 33(42.3) | |
| **Education** | Completed Primary or Less | 469(54.3) | 570 (58.1) | 101(85.6) | < .0001 | 529 (56.2) | 60(76.9) | **0.0001** |
| | Some Secondary or More | 394(45.7) | 411 (41.9) | 17(14.4) | | 412 (43.8) | 18(23.1) | |
| **Marital Status** | Single or Widowed | 558(64.7) | 610 (62.2) | 52(44.1) | < .0001 | 598 (63.5) | 40(51.3) | **0.0059** |
| | Separated or Divorced | 90(10.4) | 120 (12.2) | 30(25.4) | | 107 (11.4) | 17(21.8) | |
| | Married(Mono, Poly, Cohab) | 215(24.9) | 251 (25.6) | 36(30.5) | | 236 (25.1) | 21(26.9) | |
| **Income Quintile** | Q1 (>20%) | 176(20.4) | 211 (21.5) | 35(29.7) | 0.1799 | 196 (20.8) | 20(25.6) | 0.3103 |
| | Q2 (20–39%) | 154(17.8) | 174 (17.7) | 20(16.9) | | 170 (18.1) | 16(20.5) | |
| | Q3 (40–59%) | 200(23.2) | 227 (23.1) | 27(22.9) | | 210 (22.3) | 10(12.8) | |
| | Q4 (60–79%) | 159(18.4) | 178 (18.1) | 19(16.1) | | 174 (18.5) | 15(19.2) | |
| | Q5 (<80%) | 174(20.2) | 191 (19.5) | 17(14.4) | | 191 (20.3) | 17(21.8) | |
| **Employment Status** | No | 32(3.7) | 33(3.4) | 1(0.9) | 0.108 | 32(3.4) | 0(0.0) | 0.0834 |
| | Yes | 830(96.3) | 946 (96.6) | 116(99.1) | | 908 (96.6) | 78 (100.0) | |
| *Missing 'Employment status': n =* | | *1* | | *1* | | | *0* | |
| **Age at Sexual Debut** | ≤15 years | 407(47.5) | 466 (47.8) | 59(50.0) | **0.046** | 443 (47.4) | 36(46.8) | 0.3775 |
| | 16–18 years | 355(41.4) | 399 (40.9) | 44(37.3) | | 387 (41.4) | 32(41.6) | |
| | 19–22 years | 89(10.4) | 100 (10.3) | 11(9.3) | | 96(10.3) | 7(9.1) | |
| | ≥23 years | 6(0.7) | 10(1.0) | 4(3.4) | | 8(0.9) | 2(2.6) | |
| *Missing 'Age at sexual debut':n =* | | *6* | | *0* | | | *1* | |
| **Number of Sexual Partners in Last 3 Months** | 0–1 partners | 184(21.3) | 216 (22.0) | 32(27.1) | 0.1687 | 202 (21.5) | 18(23.4) | 0.6253 |
| | 2–5 partners | 452(52.4) | 501 (51.1) | 49(41.5) | | 489 (52.0) | 37(48.1) | |
| | 6–10 partners | 99(11.5) | 116 (11.8) | 17(14.4) | | 106 (11.3) | 7(9.1) | |
| | 11+ partners | 128(14.8) | 148 (15.1) | 20(16.9) | | 143 (15.2) | 15(19.5) | |
| *Missing 'Number of Sexual Partners in Last 3 Months': n =* | | *0* | | | | | *1* | |
| **Consistent Condom Use with Primary Sexual Partner** | No primary partner | 387(46.1) | 429 (45.1) | 42(37.5) | < .0001 | 422 (46.1) | 35(46.7) | 0.5748 |

*(Continued)*

**Table 1.** (*Continued*)

| Parameter | Category | PLWOH, n(%) Total N = 863 | Previously diagnosed HIV status, n(%) Total N = 981 | With HIV N = 118 | P-value (Previously diagnosed vs. PLWOH) | Newly diagnosed HIV status, n(%) Total N = 941 | With HIV N = 78 | P-value (Newly diagnosed vs. PLWOH |
|---|---|---|---|---|---|---|---|---|
| | Not Consistent | 379(45.1) | 420 (44.1) | 41(36.6) | | 415 (45.4) | 36(48.0) | |
| | Consistent | 74(8.8) | 103 (10.8) | 29(25.9) | | 78(8.5) | 4(5.3) | |
| *Missing 'Consistent condom use with primary sexual partners': n =* | | 23 | | 6 | | | 3 | |
| **Consistent Condom Use with Secondary Sexual Partner** | No secondary partner | 108(12.8) | 132 (13.8) | 24(21.4) | **0.0068** | 121 (13.2) | 13(17.1) | 0.5697 |
| | Not Consistent | 360(42.7) | 393 (41.2) | 33(29.5) | | 391 (42.5) | 31(40.8) | |
| | Consistent | 375(44.5) | 430 (45.0) | 55(49.1) | | 407 (44.3) | 32(42.1) | |
| *Missing 'Consistent condom use with secondary sexual partner': n =* | | 20 | | 6 | | | 2 | |
| **Transactional sex** | No | 331(39.3) | 370 (38.8) | 39(34.8) | 0.3596 | 357 (38.9) | 26(34.2) | 0.3824 |
| | Yes | 511(60.7) | 584 (61.2) | 73(65.2) | | 561 (61.1) | 50(65.8) | |
| *Missing 'exchange sex in the last 3 months': n =* | | 21 | | 6 | | | 2 | |
| **History of Alcohol Abuse** | No | 801(92.8) | 913 (93.1) | 112(94.9) | 0.3997 | 876 (93.1) | 75(96.2) | 0.2656 |
| | Yes | 62(7.2) | 68(6.9) | 6(5.1) | | 65(6.9) | 3(3.8) | |
| MSM | No | 795 (93.8) | 905 (93.7) | 110 (93.2) | 0.82 | 864 (93.5) | 69 (90.8) | 0.32 |
| | Yes | 53 (6.20) | 61 (6.3) | 8 (6.8) | | 60 (6.5) | 7 (9.2) | |
| **Anal Sex with a man** | No | 779(90.3) | 882 (89.9) | 103(87.3) | 0.3137 | 846 (89.9) | 67(85.9) | 0.22 |
| | Yes | 84(9.7) | 99(10.1) | 15(12.7) | | 95(10.1) | 11(14.1) | |
| **Drinking before Sex in last 3 months** | Never | 449(53.6) | 511 (53.8) | 62(55.9) | 0.2516 | 486 (53.3) | 37(50.0) | 0.7635 |
| | Sometimes | 355(42.4) | 403 (42.5) | 48(43.2) | | 388 (42.5) | 33(44.6) | |
| | Always | 34(4.1) | 35(3.7) | 1(0.9) | | 38(4.2) | 4(5.4) | |
| *Missing 'drinking before sex in last 3 months': n =* | | 25 | | 7 | | | 4 | |
| **Diagnosed with STI in past 3 months** | No | 841(97.5) | 955 (97.3) | 114(96.6) | 0.5939 | 918 (97.6) | 77(98.7) | 0.4876 |
| | Yes | 22(2.5) | 26(2.7) | 4(3.4) | | 23(2.4) | 1(1.3) | |

All data are presented as n (row %). Comparisons were made between groups using Pearson's Chi-square test or, in cases with small cell sizes, Fisher's exact test.

previously aware of their status—they were newly-diagnosed in our study—which highlights missed opportunities for HIV case finding in routine practice.

Undiagnosed HIV is a key obstacle to prevention efforts and poses a great threat to the gains achieved so far in epidemic control. A study in the United States showed that sexual transmission of HIV by people who did not know they were living with HIV was 3.5 times higher than transmission by people who were aware of their HIV status [20]. Another study showed an estimated 49% of transmissions were from the 20% people who were living with undiagnosed HIV [21]. Ongoing surveillance for recently acquired HIV in national testing services in countries in sub-Saharan Africa may help inform targeted testing interventions to reduce this gap in case detection [22].

One of the action steps in PEPFAR's approach to epidemic control is acceleration of optimized HIV testing and treatment strategies to reach men under 35 years. More than half of the men in this category are unaware of their status, thus fueling the epidemic in young women aged 15–24 years and young men aged 25–35 years [23]. This is consistent with our findings where there were more women compared to men in the previously-diagnosed group. This observation may be at least partially driven by the fact that women tend to interact more with healthcare facilities, for example through antenatal care visits or family planning encounters, and they also tend to accompany their young children for sick and well child visits. Frequent touch points with healthcare systems create opportunities for HIV testing that may not exist for men who don't have this frequency of healthcare encounters. Since women are disproportionately affected by HIV in sub-Sahara Africa [7, 17], they are highly targeted by existing HIV prevention programs. On the other hand, in our analysis of people with newly-diagnosed HIV, there was no significant sex difference, suggesting a need to scale up case finding to all vulnerable populations, irrespective of sex.

Both previously-diagnosed and newly-diagnosed HIV prevalence increased with age, which is expected since HIV is a chronic and irreversible diagnosis; as time at risk since sexual debut accrues, so too will new cases. After controlling for age, higher education level was associated with lower risk for HIV in both the previously-diagnosed and newly-diagnosed groups. Education is a key component in the HIV response that is highlighted in the UNAIDS fast track strategy in achieving zero infections [24]. Education provides knowledge on HIV prevention and transmission, boosts self-esteem especially among adolescents, and helps to inform decision-making in sexual and reproductive health matters [24]. Additionally, it increases the chance for gainful employment, thus reducing poverty that may be associated with increased HIV risk [25].

Coupled with suppressive ART for PLWH, changes in sexual behavior can play a key role in reducing HIV transmission. Safer sex practices can prevent HIV transmission by people who are not aware that they are living with HIV and can also prevent transmission by people who have been diagnosed with HIV but have not achieved viral suppression. In this study, condom use with primary sexual partners was seen to be higher among the previously-diagnosed group as compared to people without HIV, suggesting that this behavior may have been impacted by knowledge of their HIV status. In contrast, people with newly-diagnosed HIV did not differ in condom usage from those without HIV. Our findings corroborate those from, a population-based impact survey in Ethiopia [26] and a systemic review and meta-analysis of studies in the United States and Canada [9] that showed an association between knowledge of HIV status and more frequent condom use. This finding points to the impact of knowing one's HIV status on sexual behaviors and explains yet another pathway by which early HIV diagnosis potentially reduces transmission. This further underscores HIV testing as a key element in achieving epidemic control. However, in this study, knowledge of HIV status did not appear to influence condom use with secondary sexual partners (casual/occasional), highlighting a potential gap in HIV prevention messaging or barriers to condom use negotiation with certain types of sexual partners. Further research on this topic is needed. Such research informs policies that have contributed to significant progress in approaching epidemic control [27], but much more work need to be done to generate new evidence and translate that evidence into new HIV prevention and treatment programming.

Interestingly, participants with an age of sexual debut of 15 years or older were about twice as likely to have previously-diagnosed HIV as compared to individuals with earlier sexual debut, but there was no such difference in the analysis of newly-diagnosed HIV. This suggests more effective case finding among participants with an older age of sexual debut. Prior data suggest that individuals with a self-perceived high risk for HIV may delay their sexual debut

**Table 2. Multivariable models of factors potentially associated with previously-diagnosed and newly-diagnosed HIV.**

| | | Previously diagnosed HIV Status | | Newly diagnosed HIV Status | |
| --- | --- | --- | --- | --- | --- |
| | | —————————- | | —————————- | |
| Parameter | Category | Prevalence Ratio | P-value | Prevalence Ratio | P-value |
| | | (95%CI) | | (95%CI) | |
| **Sex** | Male | Reference | - | Reference | - |
| | Female | **2.70 (1.69, 4.28)** | **<0.0001** | 1.05 (0.65, 1.69) | 0.8427 |
| **Education** | | | | | |
| | Completed Primary or less | Reference | - | Reference | - |
| | Secondary or more education | **0.48 (0.29, 0.80)** | **0.0045** | **0.58 (0.34, 1.00)** | **0.0494** |
| **Age** | | | | | |
| | 18–24 years | Reference | - | Reference | - |
| | 25–29 years | **2.62 (1.51, 4.57)** | **0.0007** | **3.69 (1.81, 7.52)** | **0.0003** |
| | > = 30 years | **3.70 (2.13, 6.42)** | **0.0001** | **5.68 (2.62, 12.31)** | **< .0001** |
| **Marital Status** | | | | | |
| | Single or Widowed | Reference | - | Reference | - |
| | Separated or Divorced | 1.24 (0.81, 1.89) | 0.3183 | 1.06 (0.57, 1.97) | 0.8618 |
| | Married (Mono, Poly, Cohab) | 1.30 (0.82, 2.07) | 0.2663 | 0.67 (0.33, 1.34) | 0.2563 |
| **Income Quintile** | | | | | |
| | Q1 (<20%) | Reference | - | Reference | - |
| | Q2 (20–39%) | **0.63(0.40, 0.99)** | **0.0460** | 0.76 (0.4, 1.47) | 0.4174 |
| | Q3 (40–59%) | 0.86 (0.56, 1.33) | 0.4965 | **0.45 (0.21, 0.95)** | **0.0358** |
| | Q4 (60–79%) | 0.81 (0.48, 1.36) | 0.4289 | 0.64 (0.33, 1.26) | 0.1945 |
| | Q5 (>80%) | 0.52 (0.29, 0.91) | **0.0225** | 0.72 (0.38, 1.35) | 0.3046 |
| **Consistent Condom Use with Primary Partner** | | | | | |
| | Not consistent | Reference | - | Reference | - |
| | No primary Partner | 1.46 (0.89, 2.38) | 0.1291 | 1.03 (0.58, 1.85) | 0.9166 |
| | Consistent Use | **3.19 (2.09, 4.86)** | **< .0001** | 0.73 (0.25, 2.1) | 0.5606 |
| **Consistent Condom Use with Secondary Partner** | | | | | |
| | Not consistent | Reference | - | Reference | - |
| | No secondary partner | 1.49 (0.92, 2.41) | 0.1073 | 1.28 (0.66, 2.51) | 0.465 |
| | Consistent Use | 1.111 (0.74, 1.67) | 0.6103 | 1.02 (0.62, 1.65) | 0.9497 |
| **Age of Sexual Debut** | | | | | |
| | ≤15 years | Reference | - | Reference | - |
| | 16–18 years | 0.95 (0.65, 1.37) | 0.7628 | 1.07 (0.66, 1.73) | 0.776 |
| | 19–22 years | 0.85 (0.46, 1.55) | 0.5866 | 0.91 (0.43, 1.94) | 0.81 |
| | ≥23 years | 2.36 (1.03, 5.39) | **0.0423** | 2.42 (0.66, 8.91) | 0.185 |

Bold values are those considered significant at the p<0.05 level.

[28, 29], and this perception may also influence health-seeking behaviours such as engagement with HIV testing.

In this study, HIV testing history, risk behavior and sexual history were self-reported through interviews, thus introducing potential recall and social desirability biases. The study was conducted amongst individuals presenting to be assessed for eligibility to enroll in a longitudinal HIV incidence study and findings from this population may not be generalizable to others, though the HIV prevalence detected in our study was similar to that reported by other researchers studying the general population of Kisumu County [17].

In conclusion, our findings highlight critical gaps in HIV case finding that must be addressed to achieve epidemic control. Our findings also paint a picture of a mixed epidemic that varies by geographic region and population characteristics. These analyses reinforce the need for strategic expansion of HIV testing services targeting vulnerable populations by

implementing more innovative strategies in scaling up case finding irrespective of sex such as the social network strategy, door to door testing, extended clinic hours, and other person-centered approaches. Interventions uniquely targeted to engage young men for HIV testing are also needed. Our findings support the inclusion of education as a key pillar in epidemic response, thus governments should invest in education to ensure equity, affordability, and accessibility. To end the cycle of transmission, prevention efforts should also target PLWH in providing information on how to reduce transmission. Lastly, further research on sexual behaviors and health-seeking behaviors are needed to strengthen uptake of prevention interventions and adherence to treatment.

## Acknowledgments

The authors are thankful to the study participants who made this work possible. Institutions that collaborated on this work included the Kenya Medical Research Institute (KEMRI), US Army Medical Research Directorate–Africa (USMRD-A), Henry M. Jackson Foundation for the Advancement of Military Medicine (HJF), HJF Medical Research International (HJFMRI), US Military HIV Research Program (MHRP), Walter Reed Army Institute of Research (WRAIR), and the Kenya Ministry of Health. Material has been reviewed by the Walter Reed Army Institute of Research. There is no objection to its presentation and/or publication. The opinions or assertions contained herein are the private views of the author, and are not to be construed as official, or as reflecting true views of the U.S. Department of the Army, U.S. Department of Defense, or Henry M. Jackson Foundation for the Advancement of Military Medicine. The investigators have adhered to the policies for protection of human subjects as prescribed in AR 70–25. In addition to the masthead authors of this manuscript, the **RV393 Study Group** includes: Rachel Adongo, Rachel Aguttu, Hosea Akala, Michael Bondo, Erica Broach, Christine Busisa, Nate Copeland, Jessica Cowden, Mark de Souza, Leigh Anne Eller, Milicent Gogo, Zebiba Hassen, Dale Hu, Michelle Imbach, Anne Juma, Oscar Kasera, Qun Li, Margaret Mbuchi, Mark Milazzo, Kayvon Modjarrad, Eric Ngonda, Jacob Nyariro, Jew Ochola, Roseline Ohore, Thomas Okumu, Mary Omondi, Timothy Omondi, Linnah Ooro, Beatrice Orando, June Otieno, Victorine Owira, Roselyn Oyugi, Eric Rono, and Chi Tran. The Team Lead for the RV393 Study Group is John Owuoth (John.Owuoth@usamru-k.org).

## Author Contributions

**Conceptualization:** Merlin L. Robb, Julie A. Ake, Christina S. Polyak.

**Data curation:** Adam Yates, Qun Li, Zebiba Hassen, Mark Milazzo.

**Funding acquisition:** Merlin L. Robb, Julie A. Ake.

**Methodology:** Valentine Sing'oei, John Owuoth, June Otieno, Michelle Imbach.

**Project administration:** Valentine Sing'oei, Erica Broach, Tsedal Mebrahtu.

**Supervision:** Trevor A. Crowell.

**Writing – original draft:** Valentine Sing'oei, Chiaka Nwoga, Trevor A. Crowell.

**Writing – review & editing:** Chiaka Nwoga, Trevor A. Crowell.

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
