## [Decision Letter · Decision Letter 0]

15 Aug 2023

PONE-D-23-21070HIV prevalence and awareness among adults presenting for enrolment into a study of people at risk for HIV in Kisumu County, Western KenyaPLOS ONE

Dear Dr. Crowell,

Thank you for submitting your manuscript to PLOS ONE. After careful consideration, we feel that it has merit but does not fully meet PLOS ONE’s publication criteria as it currently stands. Therefore, we invite you to submit a revised version of the manuscript that addresses the points raised during the review process.

We look forward to receiving your revised manuscript.

Kind regards,

Dickens Otieno Onyango

Academic Editor

PLOS ONE

Journal Requirements:

"The authors are thankful to the study participants who made this work possible. Institutions that supported this work included the Kenya Medical Research Institute (KEMRI), US Army Medical Research Directorate – Africa (USMRD-A), Henry M. Jackson Foundation for the Advancement of Military Medicine (HJF), HJF Medical Research International (HJFMRI), US Military HIV Research Program (MHRP), Walter Reed Army Institute of Research (WRAIR), and the Kenya Ministry of Health."

"This work was supported by cooperative agreements (W81XWH-11-2-0174; W81XWH-18-2-0040) between the Henry M. Jackson Foundation for the Advancement of Military Medicine, Inc., and the U.S. Department of Defense. This research was funded, in part, by the U.S. National Institute of Allergy and Infectious Diseases.

3. One of the noted authors is a group or consortium: RV393 Study Group 

In addition to naming the author group, please list the individual authors and affiliations within this group in the acknowledgments section of your manuscript. Please also indicate clearly a lead author for this group along with a contact email address.

5. We note that Figure 1 in your submission contain map images which may be copyrighted. All PLOS content is published under the Creative Commons Attribution License (CC BY 4.0), which means that the manuscript, images, and Supporting Information files will be freely available online, and any third party is permitted to access, download, copy, distribute, and use these materials in any way, even commercially, with proper attribution. For these reasons, we cannot publish previously copyrighted maps or satellite images created using proprietary data, such as Google software (Google Maps, Street View, and Earth). For more information, see our copyright guidelines: http://journals.plos.org/plosone/s/licenses-and-copyright.

(1) You may seek permission from the original copyright holder of Figure 1 to publish the content specifically under the CC BY 4.0 license.  

Reviewers' comments:

Reviewer's Responses to Questions

**Comments to the Author**

1. Is the manuscript technically sound, and do the data support the conclusions?

Reviewer #1: Yes

Reviewer #2: Yes

2. Has the statistical analysis been performed appropriately and rigorously? 

Reviewer #1: Yes

Reviewer #2: Yes

3. Have the authors made all data underlying the findings in their manuscript fully available?

Reviewer #1: No

Reviewer #2: Yes

4. Is the manuscript presented in an intelligible fashion and written in standard English?

Reviewer #1: Yes

Reviewer #2: Yes

5. Review Comments to the Author

Reviewer #1: HIV prevalence and awareness among adults presenting for enrolment into a study of people at risk for HIV in Kisumu County, Western Kenya

Review comments:

Lines 52 -54 should be pushed up to begin at line 46 for ease of flow in HIV statistical data.

Line 171: according to table 1, newly diagnosed males stand at 41% and not 46% as indicated

Line 171: the results on age are not consistent with the data on table 1. I know median data is reported but can the same info be added on the table?

Line 216: this is true but sample size for both surveys were different, it may be unfair to conclude that the findings are similar.

Table 1: under sex, was there any data collected from the LGBTQ population? The study was recruiting key pops and hence the sex results expected should be more than just male or female. Anal sex is reported but that can mean either MSM or females having anal sex with men?

Lines 222-224: this is already reported under conclusion. There are bits of repetition here that the authors can revise.

Line 231: on vertical transmission, it would be important to break down the newly-diagnosed data into age vs sex to better understand what age group would mostly be affected here. It may be all since the study age group was 18-35

Line 248: relates to comment on sex above – did the survey question for sexual orientation so that the LGBTQ population is covered? These are the key pops at highest risk

Reviewer #2: The authors present a secondary cross-sectional analysis of a longitudinal HIV incidence cohort. They separately compare those with previous HIV diagnosis and undiagnosed infection to those not infected with HIV, focusing on identifying differences between these two groups and those not infected, respectively. They also present geographical variation in HIV prevalence using a deviance method. The manuscript is very clearly written and the analyses clearly presented. They adequately document ethical approvals and informed consent procedures. The authors should be commended for submitting such a well-developed manuscript and clearly articulated analysis.

General comments:

The results are 5 years old, potentially limiting their impact, as HIV programs have been moving quickly to newly diagnose and enroll on treatment people living with HIV in Kenya over the last few years.

The approach of comparing the two groups of interest to a reference group of not HIV infected, while reasonable, is less obvious and straightforward than just comparing the newly diagnosed to previously diagnosed. At least one of your discussion points appear to be drawing inferences from this comparison which was never presented. Perhaps this approach was taken due to limitations in statistical power with the direct comparison among a sample of 196 HIV+ respondents.

There is a recommendation to focus on young men which is not grounded in the results, at least as presented, which do not provide a combined age/sex analysis. Table 2 shows no difference by sex in new diagnosis, and increased risk of newly diagnosed infection (as well as prior diagnosis) by age, with those aged 18-24 being least likely to be newly diagnosed.

Specific comments:

line 24: The conclusion may be accurate but does not appear to be supported by the results. Given that HIV prevalence is significantly higher in women than men, and given that your reference group is HIV-, one would expect previous diagnosis in women to be higher than men even if the proportion of those HIV infected that are diagnosed is equal between sexes. It should be rewritten to avoid depending on an comparison that you avoided making in your analysis.

Line 27: I believe word 'consistent' is missing before condom use

Line 53-54: consider noting the proportion of undiagnosed HIV that is in sub-Saharan Africa, which is more relevant to your findings than just the overall burden of PLHIV.

Line 56: I don't believe Kenya is in the top 5 countries in terms of HIV prevalence, please confirm/revise.

Line 241: this statement appears to be comparing diagnosed to undiagnosed, which you did not analyze. Also, the prevalence of diagnosed HIV in women vs men when comparing to the uninfected is also in part due to disparities in HIV prevalence by sex. The conclusion that young men should be targeted because more women are previously diagnosed (compared to HIV-) does not follow. You could either add a direct comparison of diagnosed to undiagnosed or revise the sentence accordingly. At most your findings seem to show there is no longer any age or sex disparity in undiagnosed (vs HIV-) and therefore, both sexes and all age groups require careful targeting/tailoring of case finding approaches.

Line 269: confirm that reference 30 is correct both for Mozambique and Ethiopia?

Line 300: word 'as' missing before the social network

Line 413: reference 16: I believe the final KENPHIA report is now available and perhaps should be cited instead of the preliminary report

Table 1 & 2: given your interest/focus on young men, suggest including an age-sex interaction in these analyses

Figure 1. two repeated words "in in" and "region region"

6. PLOS authors have the option to publish the peer review history of their article (what does this mean?). If published, this will include your full peer review and any attached files.

Reviewer #1: No

Reviewer #2: No

---

## [Author Response · Author response to Decision Letter 0]

27 Oct 2023

EDITOR COMMENTS 

RESPONSE: Thank you for this reminder. We have made some adjustments based on the guidelines provided, thus we now believe that our documents meet the PLOS ONE style requirements. 

"The authors are thankful to the study participants who made this work possible. Institutions that supported this work included the Kenya Medical Research Institute (KEMRI), US Army Medical Research Directorate – Africa (USMRD-A), Henry M. Jackson Foundation for the Advancement of Military Medicine (HJF), HJF Medical Research International (HJFMRI), US Military HIV Research Program (MHRP), Walter Reed Army Institute of Research (WRAIR), and the Kenya Ministry of Health." 

"This work was supported by cooperative agreements (W81XWH-11-2-0174; W81XWH-18-2-0040) between the Henry M. Jackson Foundation for the Advancement of Military Medicine, Inc., and the U.S. Department of Defense. This research was funded, in part, by the U.S. National Institute of Allergy and Infectious Diseases. 

RESPONSE: Our apologies for the misunderstanding of our Acknowledgement statement. The institutions listed were collaborating institutions and did not actually provide financial support. To better clarify this point, the Acknowledgement language has replaced the term “supported” to “collaborated on” in the manuscript. The updated Acknowledgement statement can be found below. 

"The authors are thankful to the study participants who made this work possible. Institutions that collaborated on this work included the Kenya Medical Research Institute (KEMRI), US Army Medical Research Directorate – Africa (USMRD-A), Henry M. Jackson Foundation for the Advancement of Military Medicine (HJF), HJF Medical Research International (HJFMRI), US Military HIV Research Program (MHRP), Walter Reed Army Institute of Research (WRAIR), and the Kenya Ministry of Health…." 

Additionally, we have removed the Funding Statement from the manuscript as the Funding statement within the online submission form is correct. 

3. One of the noted authors is a group or consortium: RV393 Study Group 

In addition to naming the author group, please list the individual authors and affiliations within this group in the acknowledgments section of your manuscript. Please also indicate clearly a lead author for this group along with a contact email address. 

RESPONSE: Thank you for this clarification. The affiliations/institutions for the RV393 Study Group are included in the Acknowledgement section. Also, a lead author for this group (Dr. John Owuoth, the Principal Investigator) has been included. 

RESPONSE: Thank you for noting this requirement. As requested, the Ethics statement has been moved to the Methods section of the manuscript. 

5. We note that Figure 1 in your submission contain map images which may be copyrighted. All PLOS content is published under the Creative Commons Attribution License (CC BY 4.0), which means that the manuscript, images, and Supporting Information files will be freely available online, and any third party is permitted to access, download, copy, distribute, and use these materials in any way, even commercially, with proper attribution. For these reasons, we cannot publish previously copyrighted maps or satellite images created using proprietary data, such as Google software (Google Maps, Street View, and Earth). For more information, see our copyright guidelines: http://journals.plos.org/plosone/s/licenses-and-copyright. 

(1) You may seek permission from the original copyright holder of Figure 1 to publish the content specifically under the CC BY 4.0 license. 

RESPONSE: Thank you for the detailed explanation for copyrighted images. We confirm and declare that Figure 1 is not a copyrighted image. It is an original figure that was developed by the research team using ArcGIS v10.3.1 thus does not meet the criteria for copyrighted material. 

RESPONSE: Thank you for providing this comment. The reference list has been reviewed in detail, and the following 4 references have been removed as they are not as relevant to the updated information provided in the manuscript: 

Formally Reference #8 – Green SN. Seizing the moment. Sur. 2017;14(26):73–82. 

Formally Reference #11 – National Aids Control Council. Kenya Aids Strategic Framework. Ministry Health Kenya [Internet]. 2015; Available from: http://nacc.or.ke/wp-content/uploads/2015/09/KASF_Final.pdf

Formally Reference #12 – Amornkul PN, Vandenhoudt H, Nasokho P, Odhiambo F, Mwaengo D, Hightower A, et al. HIV prevalence and associated risk factors among individuals aged 13-34 years in rural Western Kenya. PLoS One. 2009;4(7). 

Formally Reference #13 – Kharsany ABM, Karim QA. HIV Infection and AIDS in Sub-Saharan Africa: Current Status, Challenges and Opportunities. Open AIDS J. 2016;10(1):34–48. 

Formally Reference #14 – Birdthistle I, Kwaro D, Shahmanesh M, Baisley K, Khagayi S, Chimbindi N, et al. Evaluating the impact of DREAMS on HIV incidence among adolescent girls and young women: A population-based cohort study in Kenya and South Africa. PLoS Med [Internet]. 2021;18(10):1–19. Available from: http://dx.doi.org/10.1371/journal.pmed.1003837

Formally Reference #22 – UNAIDS. Preliminary UNAIDS 2021 epidemiological estimates GLOBAL HIV STATISTICS. Nurs Midwifery Board Aust [Internet]. 2021;440(July):1–5. Available from: https://www.resources.trojanuv.com/wp-content/uploads/2018/06/Contaminantes-emergentes-pesticidas-Hoja-informativa-ES.pdf

Formally Reference #27 – UNAIDS. On the to end AIDS. 2016; 

REVIEWER #1 COMMENTS 

Reviewer #1:HIV prevalence and awareness among adults presenting for enrolment into a study of people at risk for HIV in Kisumu County, Western Kenya 

Review comments: 

Lines 52 -54 should be pushed up to begin at line 46 for ease of flow in HIV statistical data. 

RESPONSE: Thank you for this suggestion, we have moved these lines up for better flow. 

Line 171: according to table 1, newly diagnosed males stand at 41% and not 46% as indicated 

RESPONSE: Thank you for noting this typo. The percent of newly diagnosed males has been changed to 41% to coincide with Table 1. 

Line 171: the results on age are not consistent with the data on table 1. I know median data is reported but can the same info be added on the table? 

RESPONSE: Thank you for sharing this suggestion. The information on median data was included in the text to add contextual information to what is already presented in Table 1. The authors would thus prefer not to include the median data on the table. 

Line 216: this is true but sample size for both surveys were different, it may be unfair to conclude that the findings are similar. 

RESPONSE: Thank you for this comment. We concur that the sample size for both surveys were different. The statement has been deleted. 

Table 1: under sex, was there any data collected from the LGBTQ population? The study was recruiting key pops and hence the sex results expected should be more than just male or female. Anal sex is reported but that can mean either MSM or females having anal sex with men? 

RESPONSE: Thank you, this is a fair inquiry. The study did not directly collect sexual orientation for the LGBTQ population, although, such information could be derived by utilizing responses to other behavioral questions, if necessary. Anal sex was collected on all participants as the Screening HIV Behavior Risk Assessment Questionnaire asked “In the past 3 months, what type of sexual activity did you engage in?”, and the multi-selection options were oral sex with women, oral sex with men, penile-vaginal intercourse, anal sex with women, and anal sex with men. Therefore, as you have observed, the “Anal sex with a man” variable from Table 1 could mean either MSM or females having anal sex with men. 

Lines 222-224: this is already reported under conclusion. There are bits of repetition here that the authors can revise. 

RESPONSE: Thank you for noting the repetition. The last sentence has been deleted as it was a repetition of another statement in the conclusion. 

Line 231: on vertical transmission, it would be important to break down the newly-diagnosed data into age vs sex to better understand what age group would mostly be affected here. It may be all since the study age group was 18-35 

RESPONSE: Thank you for this comment. We agree that additional analyses would be required that our data do not support, thus we have removed the sentence stating “Undiagnosed HIV also contributes to vertical transmission of HIV, which is readily prevented if HIV is diagnosed during antenatal care” to avoid confusion. 

Line 248: relates to comment on sex above – did the survey question for sexual orientation so that the LGBTQ population is covered? These are the key pops at highest risk 

RESPONSE: Thank you again for this related comment. While we recruited high-risk populations, 'at risk' as it relates to this study was mainly for those with a high number of sexual partners. As explained in the previous comment, the study questionnaire did not specifically inquire of sexual orientation. 

REVIEWER #2 COMMENTS 

Reviewer #2:The authors present a secondary cross-sectional analysis of a longitudinal HIV incidence cohort. They separately compare those with previous HIV diagnosis and undiagnosed infection to those not infected with HIV, focusing on identifying differences between these two groups and those not infected, respectively. They also present geographical variation in HIV prevalence using a deviance method. The manuscript is very clearly written and the analyses clearly presented. They adequately document ethical approvals and informed consent procedures. The authors should be commended for submitting such a well-developed manuscript and clearly articulated analysis. 

General comments: 

The results are 5 years old, potentially limiting their impact, as HIV programs have been moving quickly to newly diagnose and enroll on treatment people living with HIV in Kenya over the last few years. 

RESPONSE: Thank you for this comment. We concur that the time factor may potentially limit the impact; however, despite the declining national prevalence in Kenya, there are few high burden areas which continue to report high HIV prevalence and incidence especially among the adolescents and young people. This study was conducted in one of the high burden counties- Kisumu County and it includes the high priority populations thus the results are potentially impactful in informing the policies on prevention. 

The approach of comparing the two groups of interest to a reference group of not HIV infected, while reasonable, is less obvious and straightforward than just comparing the newly diagnosed to previously diagnosed. At least one of your discussion points appear to be drawing inferences from this comparison which was never presented. Perhaps this approach was taken due to limitations in statistical power with the direct comparison among a sample of 196 HIV+ respondents. 

RESPONSE: 

We appreciate the Reviewer’s comment. The primary comparison of interest is between people living with HIV and PLWOH. Sub-setting participants who knew they were living with HIV at the time of screening is critical in that knowledge of ones status is likely to act as an important source of variance in behavior of PLWH. While it is of interest to note where the differences in behavior between PLWH and PLWOH are concordant or discordant between HIV knowledge groups, the central relationship of interest remains between PLWH and PLWOH. 

As the Reviewer notes, down-selecting to an analysis only between PLWH disaggregated by their knowledge status would represent a reduction in statistical power. It also would represent a significant deviation from the objective of the manuscript in identifying behaviors and risk associated with HIV prevalence as the PLWOH population is necessary to determine behaviors associated with HIV acquisition. 

There is a recommendation to focus on young men which is not grounded in the results, at least as presented, which do not provide a combined age/sex analysis. Table 2 shows no difference by sex in new diagnosis, and increased risk of newly diagnosed infection (as well as prior diagnosis) by age, with those aged 18-24 being least likely to be newly diagnosed. 

RESPONSE: Thank you for this comment. We agree that our results do not any sex difference in newly diagnosed cases. This addition was an oversight from the findings of other studies, and we have removed the statement from our conclusion. 

line 24: The conclusion may be accurate but does not appear to be supported by the results. Given that HIV prevalence is significantly higher in women than men, and given that your reference group is HIV-, one would expect previous diagnosis in women to be higher than men even if the proportion of those HIV infected that are diagnosed is equal between sexes. It should be rewritten to avoid depending on an comparison that you avoided making in your analysis. 

RESPONSE: Thank you for the comment. The prevalence of previously diagnosed HIV was higher in females but there was no significant difference in newly diagnosed HIV between males and females. We have edited the Conclusion in the abstract as follows: ‘Prevalence of newly-diagnosed HIV was high, at approximately 8% of participants, and not statistically different between genders, highlighting the need for improved HIV case finding regardless of sex. The higher prevalence of previously-diagnosed HIV in female participants may reflect higher rates of HIV testing through more encounters with the healthcare system. Higher prevalence of consistent condom use amongst those previously-diagnosed suggests behavioral change to reduce HIV transmission, a potential benefit of policies to facilitate earlier HIV diagnosis.’ 

Line 27: I believe word 'consistent' is missing before condom use 

RESPONSE: Thank you for informing us. The term “consistent” has been added as noted. 

Line 53-54: consider noting the proportion of undiagnosed HIV that is in sub-Saharan Africa, which is more relevant to your findings than just the overall burden of PLHIV. 

RESPONSE: Thank you for this suggestion. There was no available data specific to sub-Saharan Africa, but we did find data on the African region. Text was added to show undiagnosed PLWH and treatment gaps in the African region, compared to the previous sentence that presents this data globally. The following text was added to lines 52-55: ‘In the African region, 90% of PLWH knew their status, of which 82% were accessing treatment; this means 10% were unaware of their status and another 18% knew their stats but were not on ART.’ 

Line 56: I don't believe Kenya is in the top 5 countries in terms of HIV prevalence, please confirm/revise. 

RESPONSE: Thank you for identifying this error. Kenya is actually within the top 15 countries. We have edited this statement and added a new reference from the WHO: https://www.who.int/data/gho/data/indicators/indicator-details/GHO/prevalence-of-hiv-among-adults-aged-15-to-49-(-). 

Line 241: this statement appears to be comparing diagnosed to undiagnosed, which you did not analyze. Also, the prevalence of diagnosed HIV in women vs men when comparing to the uninfected is also in part due to disparities in HIV prevalence by sex. The conclusion that young men should be targeted because more women are previously diagnosed (compared to HIV-) does not follow. You could either add a direct comparison of diagnosed to undiagnosed or revise the sentence accordingly. At most your findings seem to show there is no longer any age or sex disparity in undiagnosed (vs HIV-) and therefore, both sexes and all age groups require careful targeting/tailoring of case finding approaches. 

RESPONSE: Thank you for this comment. The statement has been edited to ‘This is consistent with our findings where there were more women compared to men in the previously diagnosed group.” The statement on young men should be targeted more... Has been deleted and the conclusion on targeting both sexes in case finding approaches has been added. 

Line 269: confirm that reference 30 is correct both for Mozambique and Ethiopia? 

RESPONSE: Thank you for noting this error. The reference 30 is for Ethiopia. The national survey in Mozambique was an error. We have deleted it. 

Line 300: word 'as' missing before the social network 

RESPONSE: Thank you for informing us. The word “as” has been added as noted. 

Line 413: reference 16: I believe the final KENPHIA report is now available and perhaps should be cited instead of the preliminary report 

RESPONSE: Thank for this comment. The reference to the final KENPHIA Report has been added to replace the preliminary report. 

Table 1 & 2: given your interest/focus on young men, suggest including an age-sex interaction in these analyses 

RESPONSE: We appreciate the reviewers suggestion. However, we have reworked the discussion as it overextended the results to young men. Please see the revision as summarized in the abstract Conclusion, which is also provided here: “Prevalence of newly-diagnosed HIV was high, at approximately 8% of participants, and not statistically different between genders, highlighting the need for improved HIV case finding regardless of sex. The higher prevalence of previously-diagnosed HIV in female participants may reflect higher rates of HIV testing through more encounters with the healthcare system. Higher prevalence of consistent condom use amongst those previously-diagnosed suggests behavioral change to reduce HIV transmission, a potential benefit of policies to facilitate earlier HIV diagnosis.” 

Figure 1. two repeated words "in in" and "region region" 

RESPONSE: Thank you for identifying these errors. The Figure 1 description has been updated to remove the repeat words. An updated Figure 1 document will be uploaded in the online resubmission (no tracked changes version will be provided).

---

## [Decision Letter · Decision Letter 1]

9 Nov 2023

HIV prevalence and awareness among adults presenting for enrolment into a study of people at risk for HIV in Kisumu County, Western Kenya

PONE-D-23-21070R1

Dear Dr. Crowell,

We’re pleased to inform you that your manuscript has been judged scientifically suitable for publication and will be formally accepted for publication once it meets all outstanding technical requirements.

Kind regards,

Dickens Otieno Onyango

Academic Editor

PLOS ONE

Additional Editor Comments (optional):

Reviewers' comments:

Reviewer's Responses to Questions

**Comments to the Author**

1. If the authors have adequately addressed your comments raised in a previous round of review and you feel that this manuscript is now acceptable for publication, you may indicate that here to bypass the “Comments to the Author” section, enter your conflict of interest statement in the “Confidential to Editor” section, and submit your "Accept" recommendation.

Reviewer #1: All comments have been addressed

Reviewer #2: All comments have been addressed

2. Is the manuscript technically sound, and do the data support the conclusions?

Reviewer #1: Yes

Reviewer #2: (No Response)

3. Has the statistical analysis been performed appropriately and rigorously? 

Reviewer #1: Yes

Reviewer #2: (No Response)

4. Have the authors made all data underlying the findings in their manuscript fully available?

Reviewer #1: Yes

Reviewer #2: (No Response)

5. Is the manuscript presented in an intelligible fashion and written in standard English?

Reviewer #1: Yes

Reviewer #2: (No Response)

6. Review Comments to the Author

Reviewer #1: The authors have addressed all concerns, and the paper has been well revised. I have no further comments.

Reviewer #2: (No Response)

7. PLOS authors have the option to publish the peer review history of their article (what does this mean?). If published, this will include your full peer review and any attached files.

Reviewer #1: No

Reviewer #2: No

---

## [Editor Report · Acceptance letter]

18 Dec 2023

PONE-D-23-21070R1 

PLOS ONE

Dear Dr. Crowell, 

I'm pleased to inform you that your manuscript has been deemed suitable for publication in PLOS ONE. Congratulations! Your manuscript is now being handed over to our production team.

Kind regards, 

on behalf of

Dr. Dickens Otieno Onyango 

Academic Editor

PLOS ONE